# Investigation on 220 GHz Taper Cascaded Over-Mode Circular Waveguide TE$_{0n}$ Mode Converter

Tongbin Yang [1], Xiaotong Guan [2,3], Wenjie Fu [1,3,*], Dun Lu [1], Xuesong Yuan [1,3] and Yang Yan [1,3]

1. School of Electronic Science and Engineering, University of Electronic Science and Technology of China, Chengdu 610054, China; yangtongbin@std.uestc.edu.cn (T.Y.); ludun@std.uestc.edu.cn (D.L.); yuanxs@uestc.edu.cn (X.Y.); yanyang@uestc.edu.cn (Y.Y.)
2. School of Physics, University of Electronic Science and Technology of China, Chengdu 610054, China; guanxt@uestc.edu.cn
3. Terahertz Science and Technology Key Laboratory of Sichuan Province, University of Electronic Science and Technology of China, Chengdu 610054, China
* Correspondence: fuwenjie@uestc.edu.cn

**Abstract:** This paper proposes a taper cascaded over-mode circular waveguide TE$_{0n}$ mode converter for the millimeter and terahertz wave gyrotron. The mode converter of this structure can effectively reduce the difficulty of high frequency mode converter in fabrication. This paper verifies the feasibility of this new structure from theory, simulation, and experiment. Based on coupled wave theory calculations, three TE$_{02}$-TE$_{01}$ mode converters with lengths of 65.43 mm (4 segments), 119.3 mm (6 segments) and 136 mm (8 segments) and a TE$_{03}$-TE$_{02}$ mode converter with a length of 92 mm (8 segments) are optimized. The conversion efficiency in the frequency band 215–225 GHz is 91.8–94%, 93–95%, 95–98.78% and 95–98.44%. Because the length of the mode converter is clearly limited, this paper selects the TE$_{02}$-TE$_{01}$ mode converter with a length of 65.43 mm (4 segments) and the TE$_{03}$-TE$_{02}$ mode converter with 92 mm (8 segments) for simulation and experimental verification. In the simulation software Computer simulation technology (CST), the TE$_{02}$-TE$_{01}$ and TE$_{03}$-TE$_{02}$ mode converters and their composed TE$_{03}$-TE$_{01}$ mode converters are selected for modeling and analyzing. The simulation results and theoretical calculation results of the three mode converters only have different degrees of frequency deviation, and the frequency deviation of the 4-stage TE$_{02}$-TE$_{01}$ mode converter can be ignored; the frequency deviations of TE$_{03}$-TE$_{02}$ mode converter and TE$_{03}$-TE$_{01}$ mode converter are 2 GHz and 3 GHz, respectively. The experimental system is a field scanning system based on a vector network analyzer (VNA), which scans the input and output of the mode converter respectively. The experimental result is that when the input mode purity is 92% in TE$_{01}$ mode, the output mode TE$_{03}$ mode has a mode purity of 82%, and it has lower transmission loss. In this paper, the results from theory, simulation and experiment are in good agreement. This type of mode converter is easy to prepare, which makes it an effective alternative for high frequency curvilinear waveguide mode converter.

**Keywords:** mode converter; 220 GHz; taper

## 1. Introduction

Vacuum electronic devices (VEDs) (e.g., Traveling-Wave Tube (TWT), Backward-Wave Oscillator (BWO), Magnetron, Klystron, Gyrotron) are important high-power microwave radiation sources for scientific, industrial and military applications [1]. As the VEDs operating frequency extends to the millimeter wave and terahertz wave bands, the size of the interaction circuit shrinks, and the power capability is consequently reduced. Thus, operating at high-order mode is proposed and attempted. Gyrotron, as a fast wave VED, is successfully operated at high-order modes. Thumm et al. reported that the gyrotron operated at the highest TE$_{32,19}$ (Transverse Electric) mode [2]. However, the high-order waveguide modes are symmetrical nonlinear polarization modes, and the axial radiation is

an unsatisfactory hollow state, which is not conducive to long-distance transmission and direct application under the condition of over-mode operation. Depending on the working conditions, these modes need to be converted. There are two types of mode converters for VEDs. One is a waveguide mode converter which is usually used for higher-order volume modes ($TE_{0n}$, n > 1) [3], and its advantages are high converter efficiency, low reflectivity, and no parasitic oscillation of the device; another is quasi-optical mode converter which is used for higher-order whispering gallery modes ($TE_{mn}$, m > 1, n = 1, 2) [4] and its advantages are high power capacity, low transmission loss, and high output mode purity.

For a waveguide mode converter, the mode conversion scheme is normally as follows:

(1)　$TE_{0n}$-$TE_{01}$-$TE_{11}$-$HE_{11}$ [5,6]
(2)　$TE_{0n}$-$TE_{01}$-$TM_{11}$-$HE_{11}$ [7–9]

within which $TE_{01}$ mode has low loss at high frequencies and is suitable for long-distance transmission; HE mode is a hybrided mode of TE (Transverse Electric) mode and TM (Transverse Magnetic) mode. $HE_{11}$ includes $TE_{11}$ (84% power) and $TM_{11}$ (16% power). $HE_{11}$ mode has the characteristics of linear polarization and Gaussian distribution, and is suitable for antenna transmission and application [6].

In both conversion sequences, the mode converting from $TE_{0n}$ mode to $TE_{01}$ mode is foremost. The $TE_{0n}$-$TE_{01}$ waveguide mode converter changes its radial characteristics through radius perturbation. A common method is a corrugated circular waveguide mode converter with cyclically sinusoidal radius change [3]. At a low frequency, the sinusoidal radius change is easy to achieve. As the operating frequency increases, the sensitivity to processing errors increases and the difficulty of its implementation also increases exponentially. When the operating frequency increase up to terahertz wave region, the fabrication is extremely difficult and expensive.

In this paper, an economical gradual-radius cascaded circular waveguide mode converter is proposed as shown in Figure 1, whose fabrication is much easier than conventional type. A prototype of $TE_{03}$-$TE_{02}$-$TE_{01}$ conversion for a frequency tunable 220 GHz gyrotron [10] is designed, manufactured, and tested. The results present a good performance at 215–225 GHz and would be an expectable approach in terahertz wave applications. This paper is organized as follows. The second part introduces the theoretical calculation and simulation results of the mode converter and gives the design ideas, main formulas, and structural parameters of the design. The third part introduces the experimental process and results analysis, and Section 4 draws the conclusion.

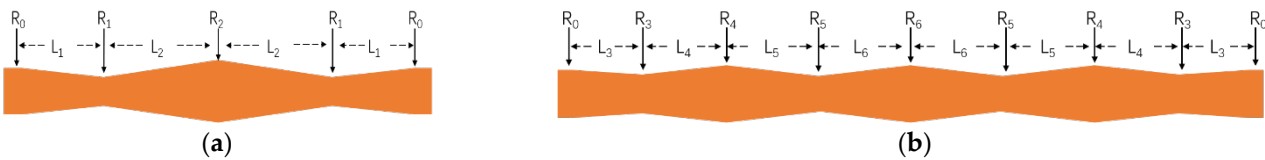

**Figure 1.** (**a**) Structure of $TE_{02}$-$TE_{01}$ mode converter; (**b**) Structure of $TE_{03}$-$TE_{02}$ mode converter.

## 2. Design Model and Simulation

### 2.1. Theoretical Calculation of Mode Converter

2.1.1. Theoretical Deduction

The inhomogeneous circular waveguide is divided into three categories: changes in transmission direction, changes in transmission medium, and changes in transmission radius. The mode converter uses the different transmission characteristics of each mode under the three changing conditions, selectively changes the three conditions, and obtains the required operating mode through mode-coupling. The common three types of circular waveguide mode converters are divided into serpentine waveguides with varying axis [11], dielectric-filled waveguides [12], and corrugated waveguides with varying transmission radius [13]. The theory of analyzing the coupling between modes in a waveguide is called

coupled wave theory [14]. The design of the mode converter with radius perturbation is to solve the boundary value problem of coupled wave ordinary differential equations, the equations are as follows:

$$\frac{dA_{mn'}^+}{dz} = -\frac{1}{2}\frac{d(\ln\gamma_{mn'})}{dz}A_{mn'}{}^- - \gamma_{mn'}A_{mn'}^+ + \sum_{+mn}A_{mn}^+ C_{(mn')(mn)}^+ + \sum_{-mn}A_{mn}^- C_{(mn')(mn)}^- \tag{1}$$

$$\frac{dA_{mn'}^-}{dz} = -\frac{1}{2}\frac{d(\ln\gamma_{mn'})}{dz}A_{mn'}{}^+ + \gamma_{mn'}A_{mn'}^- + \sum_{+mn}A_{mn}^+ C_{(mn')(mn)}^- + \sum_{-mn}A_{mn}^- C_{(mn')(mn)}^+ \tag{2}$$

where $A_{mn}^+$, $A_{mn}^-$ is the amplitude of the *mn* mode wave and the superscript indicates the direction of propagation, $\gamma_{mn}$ is the propagation constant of *mn* mode, and $\gamma_{mn} = \alpha_{mn} + j\beta_{mn}$, $\alpha_{mn}$ is the attenuation constant, $\beta_{mn}$ is phase constant, both $\alpha_{mn}$ and $\beta_{mn}$ are function of $z$, $C_{(mn')(mn)}^\pm$ is the coupling coefficient of *mn* mode and *mn'* mode in the same direction or reverse direction.

TE$_{mn}$-TE$_{mn'}$

$$C_{(mn')(mn)}^\pm = \frac{m^2\left(R_{mn'}X_{mn}^2 \pm R_{mn}X_{mn'}^2\right) \mp (R_{mn} \pm R_{mn'})X_{mn}^2 X_{mn'}^2}{(R_{mn}R_{mn'})^{1/2}(X_{mn}^2 - m)^{1/2}(X_{mn'}^2 - m)^{1/2}(X_{mn'}^2 - X_{mn}^2)}\frac{1}{a}\frac{da}{dz}(-1)^{n+n'} \tag{3}$$

TE$_{mn}$-TM$_{mn'}$

$$C_{(mn')(mn)}^\pm = \frac{m}{(R_{mn}R_{mn'})^{1/2}(X_{mn}^2 - m)^{1/3}}\frac{1}{a}\frac{da}{dz}(-1)^{n+n'+1}, \tag{4}$$

Because the transmission direction and transmission medium of the mode converter with gradual radius have not changed and the coupling coefficient of the input mode TE$_{0n}$ mode and TM mode is 0, only the coupling between TE modes can be considered in the calculation. The mode coupling of TE$_{mn}$-TE$_{mn'}$ satisfies $n' - n = 1$ and $m = 1$, so the coupling coefficient can be written as follows:

$$C_{(0n')(0n)}^\pm = \mp\frac{X_{0n}X_{0n'}(R_{0n} \pm R_{0n'})}{(R_{0n}R_{0n'})^{1/2}(X_{0n'} - X_{0n})}\frac{1}{a}\frac{da}{dz}(-1)^{n+n'} \tag{5}$$

where $X_{0n}$ ($X_{0n'}$) is the zero point value of Bessel function $J'_0(X_{0n})$ ( $J'_0(X_{0n'})$),$R_{0n} = \beta_{0n}/k_0$ and $R_{0n'} = \beta_{0n'}/k_0$, $k_0$ is the free-space wavelength, $a$ is the radius of the circular waveguide as a function of $z$.

Suppose the length of the mode converter is $L$, where $z = 0$ is the beginning of the circular waveguide and $z = L$ is the end of the circular waveguide. There is only one mode at the input port of the mode converter, the initial amplitude is 1. The output port of the waveguide does not have an aback-propagating wave. So the wave amplitude of each model at the input and output ports satisfy

$$A_{0n}^+|_{z=0} = [(1,0),(0,0),\cdots,(0,0)]^T \tag{6}$$

$$A_{0n}^-|_{z=L} = [(0,0),(0,0),\cdots,(0,0)]^T, \tag{7}$$

where the first element of the vector $\left[A_{0n}^+\right]$ is the amplitude of the input working mode, the second element is the amplitude of the output working mode, the other elements each represent a parasitic mode. Equations (6) and (7) and Equations (1) and (2) form the boundary value problem of the first-order nonlinear coupled wave differential equations. The forward wave amplitude $A_{0n}^+$ and reverse wave amplitude $A_{0n}^-$ along the $z$ can be get to solve the equations.

The center working frequency of the Gyrotron is 220 GHz. The conversion of the output mode TE$_{03}$ to TE$_{01}$ needs to be realized in two stages, namely TE$_{03}$-TE$_{02}$ and TE$_{02}$-TE$_{01}$ mode conversion. Considering the power capacity of the mode converter and the output circular waveguide radius of the Gyrotron is 5 mm, the average radius of the

mode converter's fluctuation is determined to be 5 mm. The main design indexes of the mode converter are the conversion efficiency, the working bandwidth, and the length of the converter. Editing the Matlab numerical calculation program uses the coupled wave theory and synthesizes three optimization design indicators. The optimization process is as follows: Given the initial number of cascaded segments N of the mode converter, a multi-parameter optimization model of the specific structural dimensions of each segment of the circular waveguide is established, and the main optimization goal is to maximize the conversion efficiency, taking into account the working bandwidth for optimization calculation. Change the number of cascades N and repeat the above optimization steps to obtain the most optimized design parameters. Compare the calculated results with the CST simulation results.

### 2.1.2. Calculation Results of the $TE_{02}$-$TE_{01}$ Mode Converter

The structure of the 4-segment $TE_{02}$-$TE_{01}$ mode converter is shown in Figure 1a, and the center frequency is 220 GHz. The numerical calculation takes into account factors such as multi-mode, reverse wave, and Ohmic loss, and uses the fourth-order Runge-Kutta method to perform optimization iterations, and the results of different segments are shown in Figure 2.

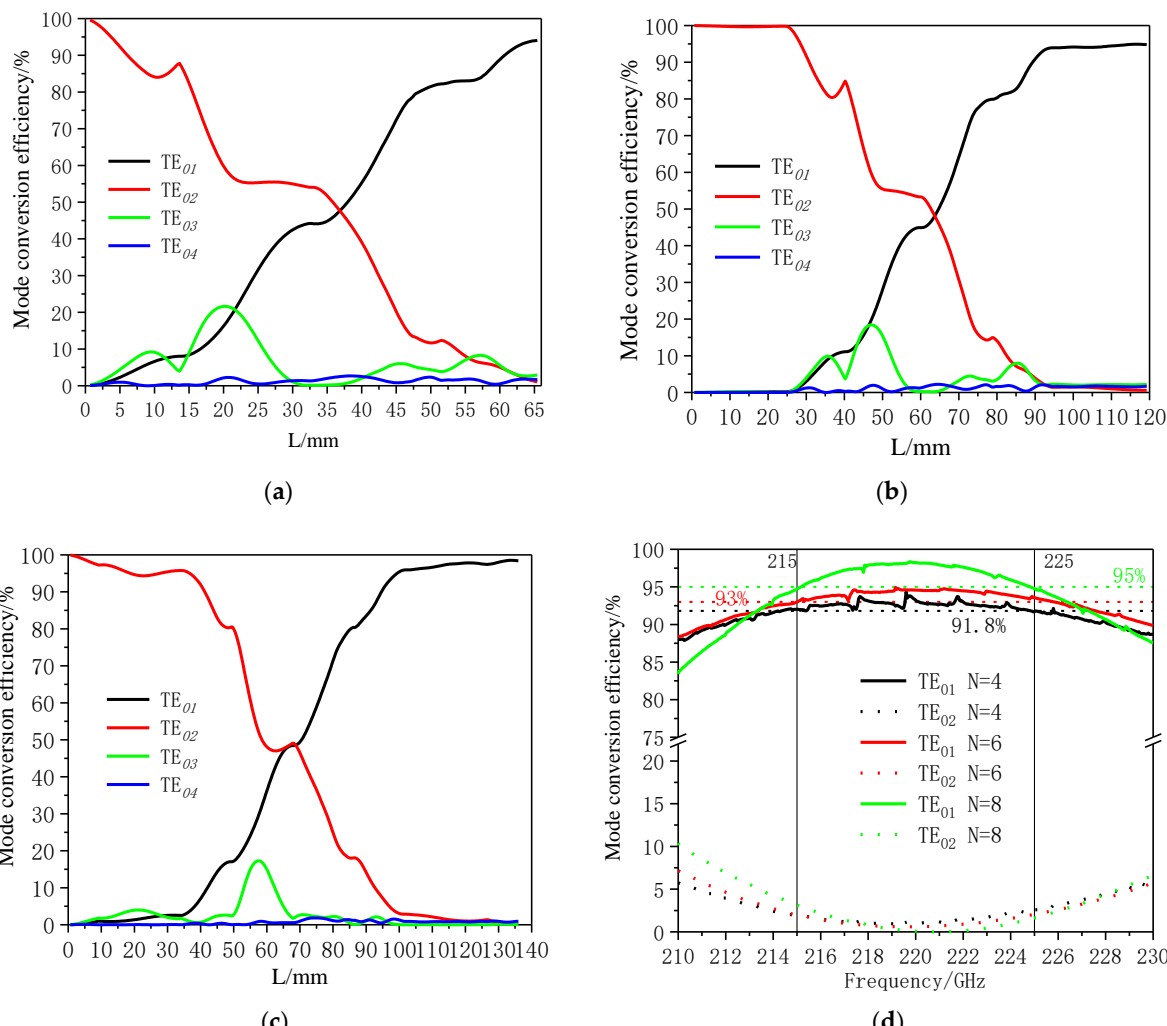

**Figure 2.** The calculation results of $TE_{02}$-$TE_{01}$ mode converter: (**a**) Mode conversion efficiency along z of a 4-segment converter; (**b**) Mode conversion efficiency along z of a 6-segment converter; (**c**) Mode conversion efficiency along z of an 8-segment converter; (**d**) Mode conversion efficiency in the different frequencies.

In the process of the numerical, multi-mode, reverse wave, ohmic loss, and other factors are taken into account. The fourth-order and fifth-order Runge-Kutta methods were used to optimize iteratively, and the converters with different segment numbers were obtained. The optimization results are shown in Figure 2. From the Figure 2a–c, it can be concluded that the parasitic modes generated during the transmission process are $TE_{03}$ and $TE_{04}$ modes. The power excited in the first half of the converter is greater, and it is effectively suppressed in the second half. The length of the 4-stage cascaded mode converter is 65.43 mm, and the mode conversion efficiency is 94% at 220 GHz. The length of the 6-segment mode converter is 119.3 mm, and the mode conversion efficiency at 220 GHz is 94.78%. The length of the 8-segment mode converter is 136 mm, and the conversion efficiency at the center frequency point is 98.78%. The Figure 2d shown that the more segments of the cascade mode changer, the longer its length and the higher the conversion efficiency. In the working frequency band 215–225 GHz, the conversion efficiency of the three mode converters are 91.8–94%, 93–95%, 95–98.78%. Comparing the length and conversion efficiency of the converter, the length of the 6-segment is almost twice the length of the 4-segment, but the overall conversion efficiency is only increased by 1–2%, and the length of the 8-segment is only 16.7 mm longer than the length of the 6-segment, the conversion efficiency has increased by 3–4%.

### 2.1.3. Calculation Results of the $TE_{03}$-$TE_{02}$ Mode Converter

The center frequency is 220 GHz, and the geometric structure of the $TE_{03}$-$TE_{02}$ mode converter is shown in Figure 1b. Using the same numerical calculation method as $TE_{02}$-$TE_{01}$ to design $TE_{03}$-$TE_{02}$ mode converter, the final result of optimization is shown in Figure 3. It can be seen from the results that the parasitic modes of the converter are $TE_{01}$ and $TE_{04}$, which are effectively suppressed during the mode conversion process. The length of the 8-segment $TE_{03}$-$TE_{02}$ mode converter is 92 mm, and the mode conversion efficiency is 98.44% at 220 GHz. The conversion efficiency in the 215–224.2 GHz frequency band is higher than 95%.

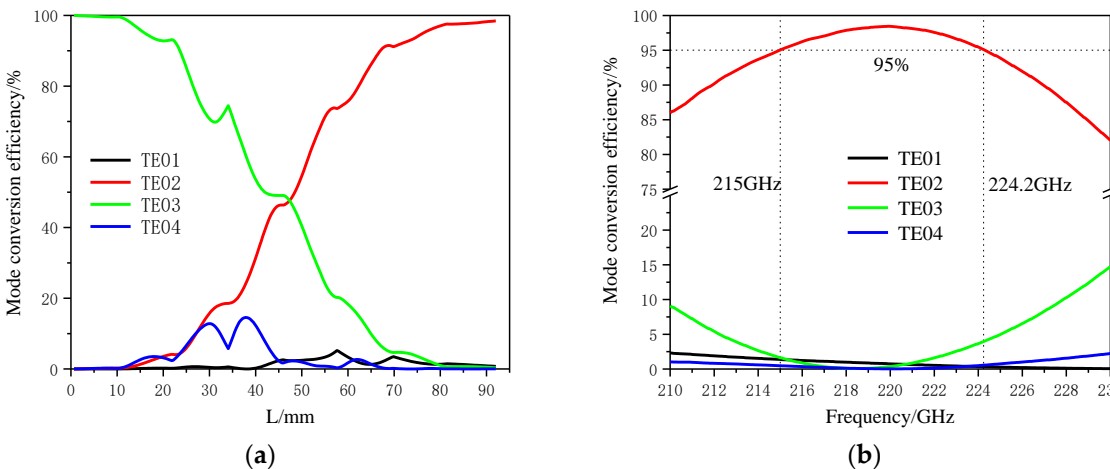

**Figure 3.** The calculation results of $TE_{03}$-$TE_{02}$ mode converter: (**a**) Mode conversion efficiency along z of the mode converter; (**b**) Mode conversion efficiency in the different frequencies.

### 2.2. Simulation Results of Mode Converters

This mode converter is a part of the output structure of the gyrotron. Due to the limitation of experimental conditions, the total length of the two modes is required to be less than 200 mm. The design scheme of the processing experiment is to take four sections of $TE_{02}$-$TE_{01}$ and six sections of $TE_{03}$-$TE_{02}$ mode converters. The two mode converters are simulated and analyzed respectively, and the $TE_{03}$-$TE_{01}$ mode converter composed of the two mode converters is analyzed and the calculation and simulation are also compared.

The mode generator in the experiment is a mode converter ($TE_{10}^{\square}$ to $TE_{01}°$) that converts from a rectangular waveguide $TE_{10}$ mode to a round waveguide $TE_{01}$ wave, and the mode converter is a reversible symmetrical two-port device. Therefore, the $TE_{01}$ mode is used as the input mode in the comparison between simulation and calculation of the $TE_{03}$-$TE_{01}$ mode converter. The structural dimensions of the calculated and simulated mode converters are shown in Tables 1 and 2.

**Table 1.** The structure size of $TE_{02}$-$TE_{01}$ mode converter.

|  | R0 | R1 | R2 | L1 | L2 |
|---|---|---|---|---|---|
| Calculation (mm) | 5 | 3.79 | 6.14 | 13.9 | 19.3 |
| Simulation (mm) | 5 | 3.79 | 6.14 | 13.9 | 19.3 |

**Table 2.** The structure size of $TE_{03}$-$TE_{02}$ mode converter.

|  | R3 | R4 | R5 | R6 | L3 | L4 | L5 | L6 |
|---|---|---|---|---|---|---|---|---|
| Calculation (mm) | 4.87 | 5.32 | 4.5 | 5.54 | 9.3 | 11 | 12.2 | 11.8 |
| Simulation (mm) | 4.87 | 5.32 | 4.5 | 5.54 | 9.3 | 11 | 12.2 | 11.8 |

Perform simulation verification in the simulation software CST [15] with the dimensions in the table, and compare it with the calculated result as shown in Figure 4. The size of the mode converter calculated by the optimization is modeled separately in the three-dimensional simulation software CST. The simulation and theoretical calculation results of $TE_{02}$-$TE_{01}$ and $TE_{03}$-$TE_{02}$ mode converters are shown in Figure 4a,b. The $S_{21}$ parameters of $TE_{01}$, $TE_{02}$, $TE_{03}$ and $TE_{04}$ in the simulation results are converted into the power content of the simulation output port, and compared with the power content of each mode output port calculated theoretically. The simulation results of the four-segment $TE_{02}$-$TE_{01}$ mode converter are basically the same than the theoretical calculation results. The conversion efficiency in the working frequency band is above 91%. In the simulation results of $TE_{03}$-$TE_{02}$ mode converter, the frequency band where the conversion efficiency of $TE_{03}$ to $TE_{02}$ remains above 95% is 217–227 GHz. Compared with the theoretical calculation result, there is a frequency offset of 2 GHz. The comparison results of other modes can also support this conclusion.

The content of each mode in the transmission direction of the $TE_{01}$-$TE_{03}$ mode converter composed of two mode converters in the theoretical calculation at the operating frequency of 220 GHz is shown in Figure 4c. At the output port of the mode converter, the mode content of $TE_{03}$ is 90%. In the simulation results, the change rule of the electric field transformation of the $TE_{01}$-$TE_{03}$ mode converter in the transmission direction conforms to the trend of each mode transformation calculated in theory. In the working frequency band 215–225 GHz, the theoretically calculated conversion efficiency is 83.8–90%. The simulation result has a frequency deviation of 3 GHz compared with the calculation result. In the frequency band 218–230 GHz, the mode conversion efficiency is 81.2–90%. The results are shown in Figure 4d.

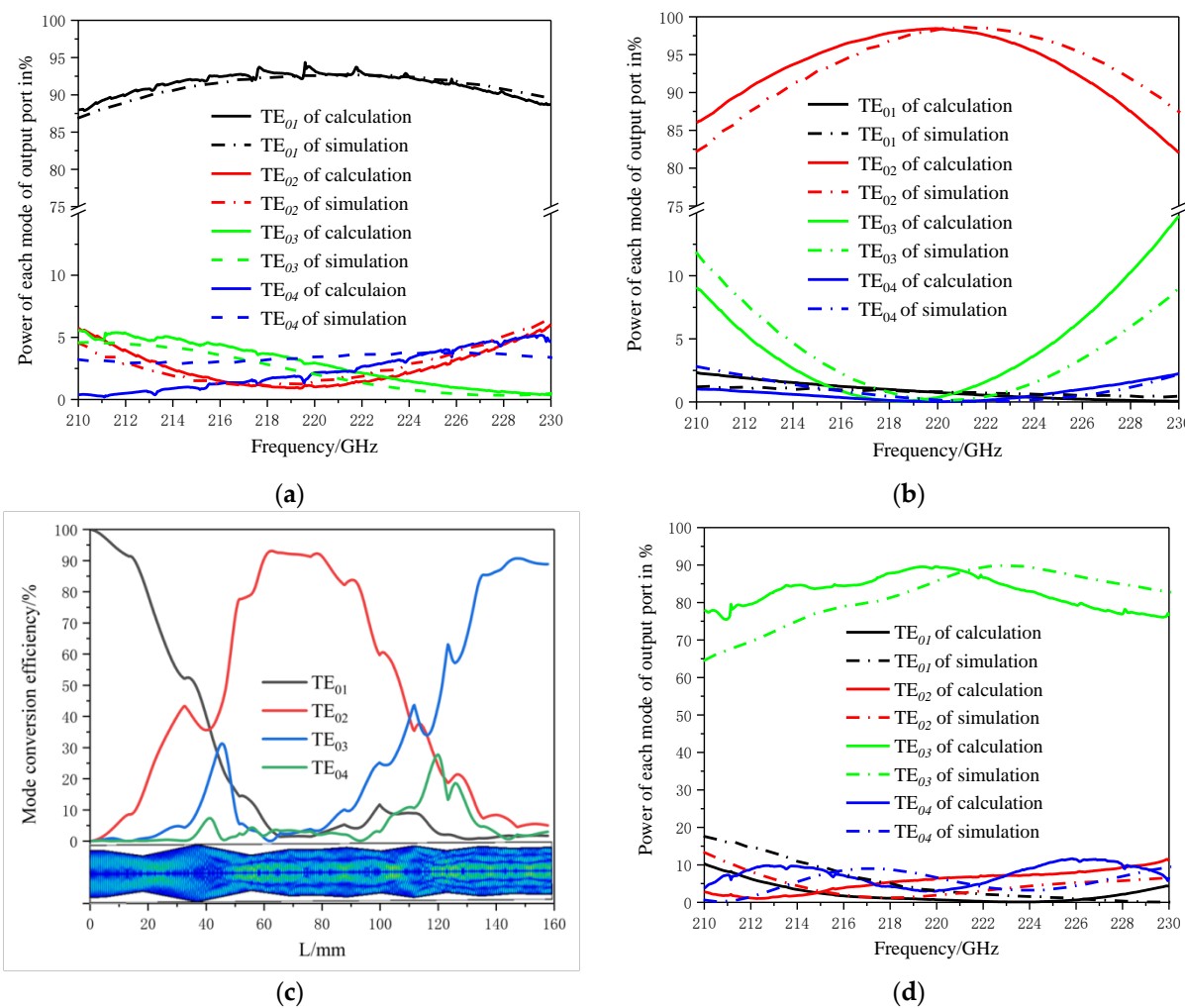

**Figure 4.** The calculation and simulation results of (**a**) 4-segment $TE_{02}$-$TE_{01}$ mode conversion efficiency. (**b**) $TE_{03}$-$TE_{02}$ mode conversion efficiency. (**c**) Calculation mode conversion efficiency and simulation E-Field of the $TE_{03}$-$TE_{01}$ mode converter along the z direction. (**d**) $TE_{03}$-$TE_{01}$ mode conversion efficiency.

## 3. Experimental Demonstration

The reversibility of the mode conversion can be used to measure the mode converter from the fundamental mode of the waveguide to the higher order mode. By measuring the $S_{21}$ of the two groups of mode converters symmetrically connected, the conversion efficiency of the mode converter and the purity of the output port mode can be calculated. However, this mode converter measurement method is not suitable for the measurement of high frequency over-mode waveguide mode converters. The over-mode waveguide mode converter and the adapter from the standard waveguide to the over-mode waveguide connected at both ends will form a resonant cavity, making it easy to excite spurious modes and generate resonance. The $S_{21}$ of the results cannot calculate the performance of the mode converter.

The test system in this paper is shown in Figure 5a. The 1-port spread spectrum module of the Vector Network Analyzer (VNA) and the device under test connected to it, are fixed on the optical platform. The 2-port spread spectrum module of the VNA and the rectangular waveguide probe connected to it are fixed on the 2D displacement platform (*X*-axis and *Y*-axis). One of the electric field components of the surface where the probe port is located can be tested through the 2D displacement platform scanning controlled by computer. The position of 1-port probe and the 2D platform remains unchanged and

the 2-port probe is rotated 90 degrees along the Z-axis and then scanned through the 2D platform remains. Another electric field component of the surface can be tested. This kind of test scheme can measure the electric field component of the tested plane in the working frequency band of the vector network analyzer spread spectrum module. In the experiment of this article, first measure the two electric field components of the input of the mode converter ($E_{IN-X}$ and $E_{IN-Y}$).

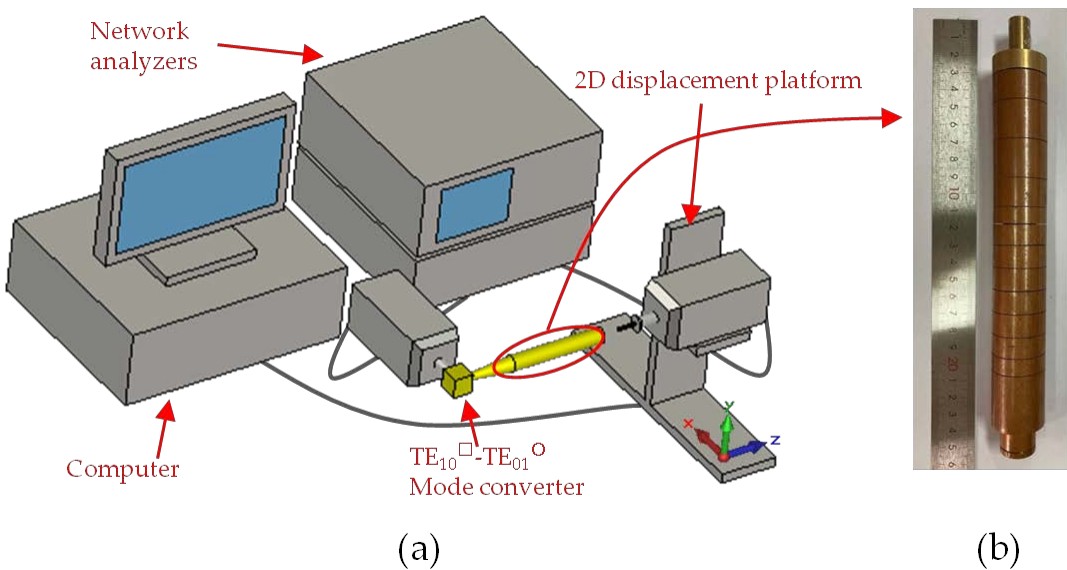

(a)                                                                            (b)

**Figure 5.** (**a**) The test systems of the converter and (**b**) The $TE_{03}$-$TE_{01}$ mode converter after brazing.

The mode converter designed in this paper is composed of two mode converters $TE_{02}$-$TE_{01}$ and $TE_{03}$-$TE_{02}$. These mode converters are segmented cascade structure. To ensure the complete connection of each component during the test, the two mode converters are integrally welded. The tested $TE_{03}$-$TE_{01}$ mode converter is composed of $TE_{03}$-$TE_{02}$ and $TE_{02}$-$TE_{01}$ mode converters. The experimental system and the mode converter after welding are shown in Figure 5. The connecting parts between the mode converter and the VNA output waveguide are a $TE_{10}^{\square}$ to $TE_{01}^{\circ}$ mode converter and a circular waveguide adapter from 1.8 mm to 10 mm in diameter. The VNA in this system is made by the 41st Institute of China Electronics Technology Group Corporation (CETC).

Figure 6 is the experimental results of electric field component at input port of the mode converter. The test results of the electric field components in the X and Y directions at the 220 GHz input port are shown in Figure 6a,b. The blue dotted line in the figure is the ideal distribution of the corresponding electric field component to $TE_{01}$ mode. It can be concluded from the comparison of the two field distribution that the two electric field components of the input port have a higher degree of matching with the electric field components of the standard $TE_{01}$ mode.

Figure 7 is the experimental results of electric field component at output port of the mode converter. The electric field at the output port of the mode converter is tested in the X and Y directions. The results at the operating frequency of 220 GHz are shown in Figure 7a,b. It can be seen from the results in the figure that the mode of the output port is $TE_{03}$ mode. Comparing the measured electric field with the ideal electric field distribution, it can be seen that the radius of the outmost peak ring of the $TE_{03}$ mode is larger than the ideal value. The position of measuring the electric field distribution is a short distance away from the opening of the waveguide. The electric field distribution tested is the result of electromagnetic wave transmission after a certain distance in the air. Due to the large diffraction of $TE_{03}$ mode in the air, the radius of the most ring is larger than the waveguide radius in the test results.

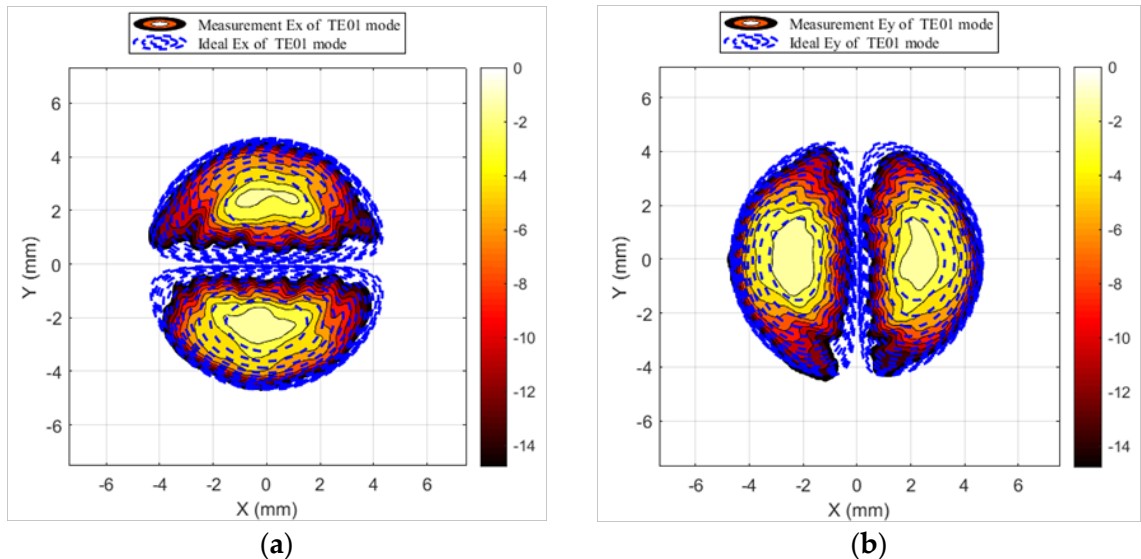

**Figure 6.** (**a**) The test results of $E_x$ of $TE_{01}$ mode; (**b**) the test results of $E_y$ of $TE_{01}$ mode.

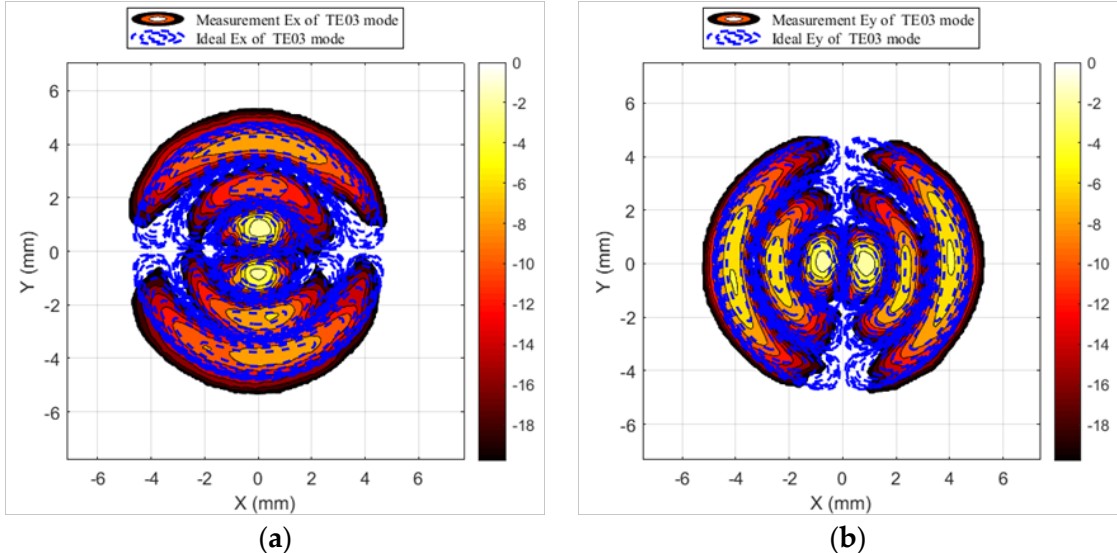

**Figure 7.** (**a**) The test results of $E_x$ of $TE_{03}$ mode; (**b**) the test results of $E_Y$ of $TE_{03}$ mode.

In order to study the performance of the tested mode converter more clearly, some frequency points of 215–225 GHz in the working bandwidth of the mode converter are selected for electric field scanning. Formula (8) can be used to calculate the mode purity of the input and output ports of the mode converter [16]. The calculation results of the purity of the electric field component measured by the input and output ports in the working frequency band is shown in Figure 8a.

$$\eta = \frac{\iint_s f(x,y) \bullet \psi^*(x,y) ds \bullet \iint_s f^*(x,y) \bullet \psi(x,y) ds}{\iint_s f(x,y) \bullet f^*(x,y) ds \bullet \iint_s \psi(x,y) \bullet \psi^*(x,y) ds} \tag{8}$$

where $f(x,y)$ represents the measured electric field distribution of the input and the output the mode converter, $\psi(x,y)$ is the ideal field distribution of the correspondence mode, the star (*) means the complex conjugation.

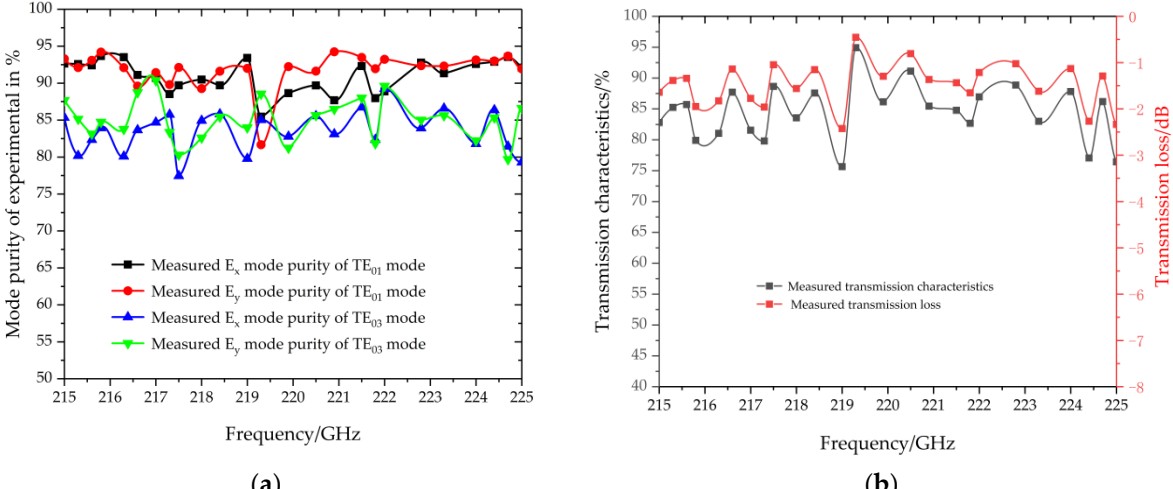

**Figure 8.** (**a**) The calculation result of the mode purity of the electric field component at the input and output of the mode converter, and (**b**) The experimental results of transmission loss of mode converter.

It can be seen from Figure 8a that at the input of the mode converter, the mode purity measured from the two electric field components is basically the same. The $TE_{01}$ mode purity at the input port of the mode converter is maintained between 90–95% in the operating frequency band, except a small band nearly 215 GHz, the mode purity is reduced to 83%. The calculation results of the mode purity corresponding to the two electric field components of the $TE_{03}$ mode measured at the output port are also basically the same. Mode purity of the $TE_{03}$ remains at about 82% in the entire operating frequency band. The mode purity curves of the electric field components in the X direction and the Y direction have good consistency at the input and output ends, indicating that the scanning planes in the two directions are basically parallel, and the uniformity of the scanning field value is good, and it shows the experimental data is reliable.

Processing the scanned field value data can calculate the transmission loss of the mode converter. Integrate the measured two electric field components on the measured plane to get the power of the two electric field components ($P_{IN-X}$ and $P_{IN-Y}$). Add the ingrate two components together, and express the sum as the input power of the mode converter as $P_{IN}$ ($P_{IN} = P_{IN-X} + P_{IN-Y}$). The output power of the mode converter is measured as $P_{OUT-X}$, $P_{OUT-Y}$ and $P_{OUT}$ ($P_{OUT} = P_{OUT-X} + P_{OUT-Y}$) in same measured method. The $S_{21}$ of the mode converter can be calculation by the calculation formula of transmission loss $S_{21} = P_{OUT}/P_{IN}$. The calculation results are shown in Figure 8b. The transmission characteristic of the mode converter is 82% in the operating frequency band and 75% in the vicinity of 219 GHz. At 219 GHz, the purity of the input mode is low, which increases the transmission loss of the millimeter wave signal.

In general, the results of the experiment show that the cascade mode converter outputs $TE_{03}$ mode with 82% mode purity when the input mode is 92% pure $TE_{01}$ mode. It is basically in line with the mode conversion efficiency of the theoretical and simulation results, and has lower transmission loss. It fully meets the requirements as a supporting device for gyrotron output.

## 4. Conclusions

In this paper, a taper cascaded over-mode circular waveguide $TE_{03}$-$TE_{01}$ mode converter for a 220 GHz gyrotron has been presented. Through the calculation of coupled wave theory, three different lengths of $TE_{02}$-$TE_{01}$ mode converters of 65.43 mm (4 segments), 119.3 mm (6 segments) and 136 mm (8 segments) are optimized, the mode conversion efficiencies of these mode converters are 91.8–94%, 93–95%, and 95–98.78%, in the design frequency band 215–225 GHz. According to the same optimization method, the $TE_{03}$-$TE_{02}$ mode converter is designed with a conversion efficiency higher than 95% in the operating

frequency band and a conversion efficiency of 98.44% at 220 GHz. Its length is 92 mm (8 segments). Because the length of the mode converter is clearly limited, this paper selects the $TE_{02}$-$TE_{01}$ mode converter with a length of 65.43 mm (4 segments) and the $TE_{03}$-$TE_{02}$ mode converter with 92 mm (8 segments) for simulation and experimental verification. 3D simulation software was used to model and simulate the two converters and the $TE_{03}$-$TE_{01}$ mode converter composed of them. The simulation result curves of the three mode converters are in good agreement with the theoretical calculation results, and there are only varying degrees of frequency deviation. The frequency deviation of the 4-stage $TE_{02}$-$TE_{01}$ mode converter can be ignored. The frequency deviations of the $TE_{03}$-$TE_{02}$ mode converter and the $TE_{03}$-$TE_{01}$ mode converter are 2 GHz and 3 GHz. In this paper, the mode of the input and output ports of the mode converter is measured by means of electric field scanning. When the input mode purity is 92% in $TE_{01}$ mode, the mode purity of $TE_{03}$ mode output of the mode converter is 82%, and the transmission loss of the measured mode converter is low. The measurement results further verify the correctness of the theoretical and simulation results, and the prepared mode converter meets the experimental requirements of our gyrotron. In general, this paper verifies the feasibility of the taper cascaded over-mode circular waveguide mode converter from three aspects of theory, simulation, and experiment. This type of mode converter is easy to prepare, which makes it an effective alternative for high frequency curvilinear waveguide mode converter.

**Author Contributions:** T.Y. and W.F. contributed to the overall study design, analysis, computer simulation, and writing of the manuscript. X.G., D.L., X.Y. and Y.Y. provided technical support and revised the manuscript. All authors have read and agreed to the published version of the manuscript.

**Funding:** This work was supported in part by National Key Research and Development Program of China under 2019YFA0210202, in part by the National Natural Science Foundation of China under Grant 61971097 and 6201101342, and in part by the Terahertz Science and Technology Key Laboratory of Sichuan Province Foundation under Grant THZSC201801.

**Institutional Review Board Statement:** Ethical review and approval were waived for this study, due to this study not involving humans or animals.

**Informed Consent Statement:** Informed consent was obtained from all subjects involved in the study.

**Data Availability Statement:** Data sharing not applicable.

**Acknowledgments:** The authors gratefully acknowledge Yin Huang and Weirong Deng for their kind assistance on engineering design and assembling

**Conflicts of Interest:** The authors declare no conflict of interest.

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
