# Peer review of "Investigation on 220 GHz Taper Cascaded Over-Mode Circular Waveguide TE0n Mode Converter"

_electronics, doi:10.3390/electronics10020103_

Round 1

Reviewer 1 Report

Yang et al. presented an investigation on cascaded tapered circular waveguides as TE0n mode converters for 220 GHz millimeter wave. In particular, they studied TE02-TE01 and TE03-TE02 mode converters with various cascaded levels. The cascaded tapered configuration is proposed to alleviate the extraordinary fabrication difficulties for their corrugated sinusoidal counterparts at this high frequency (and thus small spatial dimension). The authors have conducted theoretical calculations, numerical simulations, and measurements on their proposed designs. They obtained good agreements among theory, simulation and measurement. They have thus presented a well-supported demonstration of cascaded tapered circular waveguides as TE0n mode converters. I would thus recommend acceptance of this manuscript for the publication in Electronics, after addressing the following minor revision comments.

Page 4, line 115, “namely, TE03-TE01 and TE02-TE01 mode conversion.” should be: “namely, TE03-TE02 and TE02-TE01 mode conversion.”

Figures 6 and 7, “X-electric Field” should be called “Ex” instead; “Y-electric Field” should be called “Ey” instead. These names should be changed in the main text accordingly.

Page 2, reference [2] should describe “corrugated circular waveguide mode converter with cyclically sinusoidal radius change” in detail. However, the current reference [2] does not do the job. A more appropriate reference is needed.

Lastly, there are some English modification suggestions.

Page 2, line 49, “The mode converters for VEDs are two types.” would better be: “There are two types of mode converters for VEDs.”

Page 2, line 58, “which” would better be: “within which”.

Page 2, line 69, “which” would better be: “whose”.

Page 2, line 71, “would to be” would better be: “would be”.

Author Response

Page 4, line 115, “namely, TE03-TE01 and TE02-TE01 mode conversion.” should be: “namely, TE03-TE02 and TE02-TE01 mode conversion.”

RE

The error in the text have been corrected

Figures 6 and 7, “X-electric Field” should be called “Ex” instead; “Y-electric Field” should be called “Ey” instead. These names should be changed in the main text accordingly.

RE

These names have been changed in the text, figures and figure names.

Page 2, reference [2] should describe “corrugated circular waveguide mode converter with cyclically sinusoidal radius change” in detail. However, the current reference [2] does not do the job. A more appropriate reference is needed.

RE

This is an incorrect reference mark and has been modified.

Lastly, there are some English modification suggestions.

Page 2, line 49, “The mode converters for VEDs are two types.” would better be: “There are two types of mode converters for VEDs.”

Page 2, line 58, “which” would better be: “within which”.

Page 2, line 69, “which” would better be: “whose”.

Page 2, line 71, “would to be” would better be: “would be”.

RE

These English mistakes have been corrected

Reviewer 2 Report

Review of Electronics 1068809 Investigation on 220 GHz Taper Cascaded Over-Mode 2 Circular Waveguide TE0n Mode Converter by Yang et al.

Authors numerically solve differential equations to obtain conversion efficiency from higher order to lower order mode in a circular waveguide with different number of periods of radius variation.  Results are confirmed by simulation using commercial CST 3D electrodynamic simulation software.  The results are technologically significant.

The reviewer has only general knowledge of waveguide theory and practice, so most comments here regard mechanical deficiencies in the paper.

TE, TM, and TEM are standard designations in any undergraduate E&M treatment of waveguides, but “HE” is not.  To broaden the potential audience, please define the acronym and give a reference where these modes are described.  Here is what I copied from an online textbook:  “When the horizontal electric field component dominates the modes are sometimes called HEpq modes or TEpq modes (with a slight abuse of terminology). When the vertical electric field component dominates the modes are called EHpq modes or TMpq modes (again with a slight abuse of terminology). The two subscripts p and q indicate the number of nodes the dominant electric field component has in the horizontal and vertical directions, respectively.”

The meaning of the asterisk in the differential equations should be explained.  It conventionally means complex conjugate, and it is so defined after Eq. 8.  But in Eqs 1 etc it seems to be applied to the mode index, usually an integer, so that doesn’t make sense.  Please clarify.

Give a reference for the CST software with information on who makes it, and their address or web address.

Line 178.  Errors on figure call-outs.

The introduction states that a “prototype of TE03-TE02-TE01 conversion” device is being designed.  The first paragraph of section 2.2 says “the TE03-TE01 mode converter composed of the two mode converters is analyzed and the calculation and simulation are also compared.  However, Figs. 4c,d gives results for the reverse conversion, without explanation.  Why was that done? The answer seems to be given in the first sentence of section 3, but that clarification should come before the results of Fig. 4 are presented.

Line 226.  Please clarify the meaning of the circle and square superscripts on the mode designation.

Figs. 6a and 7a do not add anything meaningful and should be removed.  If the authors decide to keep them, please crop out all extraneous details in the background of their lab.

Author Response

TE, TM, and TEM are standard designations in any undergraduate E&M treatment of waveguides, but “HE” is not.  To broaden the potential audience, please define the acronym and give a reference where these modes are described.  Here is what I copied from an online textbook:  “When the horizontal electric field component dominates the modes are sometimes called HEpq modes or TEpq modes (with a slight abuse of terminology). When the vertical electric field component dominates the modes are called EHpq modes or TMpq modes (again with a slight abuse of terminology). The two subscripts p and q indicate the number of nodes the dominant electric field component has in the horizontal and vertical directions, respectively.”

RE

Added preliminary explanations to TE (45 lines), TM and HE modes (59-60 lines) in the text

The meaning of the asterisk in the differential equations should be explained.  It conventionally means complex conjugate, and it is so defined after Eq. 8.  But in Eqs 1 etc it seems to be applied to the mode index, usually an integer, so that doesn’t make sense.  Please clarify.

RE

In order to avoid the misunderstanding caused by the star (*) in the formulas, the formulas (1-5) in the text are modified. In the modified formula, only formula 8 contain a star (*), which represents the complex conjugate.

Give a reference for the CST software with information on who makes it, and their address or web address.

RE

Added reference to line 180, stating the company and website of CST software

Line 178.  Errors on figure call-outs.

RE

The errors in the text have been corrected.

The introduction states that a “prototype of TE03-TE02-TE01 conversion” device is being designed.  The first paragraph of section 2.2 says “the TE03-TE01 mode converter composed of the two mode converters is analyzed and the calculation and simulation are also compared.  However, Figs. 4c,d gives results for the reverse conversion, without explanation.  Why was that done? The answer seems to be given in the first sentence of section 3, but that clarification should come before the results of Fig. 4 are presented.

RE

The reason of the Figs. 4c,d gives results for the reverse conversion is "Because the mode generator in the experiment is a mode converter (TE10□ to TE01○) that converters from a rectangular waveguide TE10 mode to a round waveguide TE01 wave shown in Figure 5. And the mode converter is a reversible symmetrical two-port device. Therefore, the TE01 mode is used as the input mode in the comparison between simulation and calculation of the TE03-TE01 mode converter." Added in the text 171-175 lines.

Line 226.  Please clarify the meaning of the circle and square superscripts on the mode designation.

RE

There are instructions in the added content of lines 172-173 in the text.

Figs. 6a and 7a do not add anything meaningful and should be removed.  If the authors decide to keep them, please crop out all extraneous details in the background of their lab.

RE

Figs 6a and 7a have been deleted.
